# Free Fatty Acids from Cow Urine DMSO Fraction Induce Cell Death in Breast Cancer Cells without Affecting Normal GMSCs

**DOI:** 10.3390/biomedicines11030889

**Published:** 2023-03-13

**Authors:** Ajay Kumar Raj, Vidhi Upadhyay, Kiran Bharat Lokhande, K. Venkateswara Swamy, Ramesh Ramchandra Bhonde, Sachin C. Sarode, Nilesh Kumar Sharma

**Affiliations:** 1Cancer and Translational Research Lab, Dr. D. Y. Patil Biotechnology & Bioinformatics Institute, Dr. D. Y. Patil Vidyapeeth, Pimpri, Pune 411033, Maharashtra, India; 2Bioinformatics Research Laboratory, Dr. D. Y. Patil Biotechnology and Bioinformatics Institute, Dr. D. Y. Patil Vidyapeeth, Pimpri, Pune 411033, Maharashtra, India; 3MIT-School of Bioengineering Sciences & Research, MIT-Art, Design and Technology University, Pune 412201, Maharashtra, India; 4Regenerative Medicine Laboratory, Dr. D. Y. Patil Dental College and Hospital, Dr. Y. Patil Vidyapeeth, Pimpri, Pune 411018, Maharashtra, India; 5Department of Oral Pathology and Microbiology, Dr. D. Y. Patil Dental College and Hospital, Dr. D. Y. Patil Vidyapeeth, Pimpri, Pune 411018, Maharashtra, India

**Keywords:** metabolites, cancer, apoptosis, cytotoxicity, free fatty acids, molecular dynamics, simulations, epigenetic enzyme

## Abstract

Objective: The objective of this study was to explore the biological relevance of free fatty acids derived from cow urine DMSO fraction (CUDF) by employing in vitro and in silico approaches. Background: Metabolic heterogeneity at the intra- and intercellular levels contributes to the metabolic plasticity of cancer cells during drug-induced response. Free fatty acid (FFA) availability at intra- and intercellular levels is related to tumor heterogeneity at interpatient and xeno-heterogeneity levels. Methods: We collected fresh urine from healthy cows and subjected it to fractionation in DMSO using drying, vortexing, and centrifugation. Finally, the sterile filtrate of cow urine DMSO fraction (CUDF) was evaluated for antiproliferative and proapoptotic effects in MCF-7 and ZR-75-1 breast cancer cells using routine cell-based assays. Intracellular metabolites were studied with the help of a novel in-house vertical tube gel electrophoresis (VTGE) method to reveal the nature of CUDF components in MCF-7 cells. Identified intracellular FFAs were studied for their molecular interactions with targeted receptor histone deacetylase (HDAC) using molecular docking and molecular dynamics (MD) simulations. Results: CUDF showed a significant reduction in cell viability and cell death in MCF-7 and ZR-75-1 breast cancer cells. Interestingly, FFAs tetracosanedioic acid, 13Z-docosenoic acid (erucic acid), nervonic acid, 3-hydroxy-tetradecanoic acid, and 3-hydroxcapric acid were found inside the treated MCF-7 cancer cells. These FFAs, including tetracosanedioic acid, indicated a specific affinity to HDAC at their inhibitory sites, similar to trichostatin A, a known inhibitor. Conclusions: This study reports on FFAs derived from CUDF as potential antiproliferative and pro-cell death agents against breast cancer cells. MD simulations hinted at tetracosanedioic acid and other FFAs as inhibitors of HDAC that could explain the observed effects of FFAs in cancer cells.

## 1. Introduction

Cancer is a leading cause of death. Breast cancer kills 6.1 million women yearly due to the modern lifestyle and habitat environment [1]. The most challenging cancer issue is its multifactorial nature, which is contributed to by genetic, epigenetic, and environmental factors. Among environmental components, the dietary component is a significant factor that shapes various attributes of cancer, including initiation, growth, progression, invasiveness, and drug resistance [2,3,4,5].

In cancer, challenges are observed in the context of drug resistance, life-threatening side effects, and recurrence of secondary tumors [4,6]. Susceptibility and resistance to cancer types are linked with metabolic reprogramming within an organism, including the contributory role of gut microbiotas [7,8,9,10]. Dietary components are metabolized within the physiological settings of an organism’s gut microbial fermentation. A distinct set of metabolites from various sources is one of the reasons behind the susceptibility and resistance to cancer types that may explain the basis of intra-, inter-, and xeno-tumor heterogeneity [10,11,12,13]. Interestingly, compared to humans, cows and other ruminants show a rare occurrence of mammary cancer [14,15].

Among several classes of metabolites, free fatty acids (FFAs) derived from dietary and gut microbiotas are active components of cellular metabolic energy and reprogramming [16,17,18,19,20,21,22,23]. However, FFAs are suggested to play a protective role in chronic diseases such as cancer, hepatitis, and autoimmune diseases [24,25,26,27,28]. The important lesson from this observation is that a fiber-rich diet of vegetables, fruits, and berries supports the balance of the gut microbiota. Limited findings have been reported on the potential of FFAs as cytostatic and apoptosis inducers in cancer cells [24,25,26,27,28,29,30,31,32].

Histone deacetylase (HDAC) is a crucial regulatory epigenetic enzyme that is overexpressed in most cancer types [33,34,35,36,37,38]. Recently, pharmacological and dietary inhibitors of HDAC have been explored for their anticancer effects [33,34,35,36,37,38]. At the molecular level, FFAs have been reported to act as inhibitors of pro-proliferation proteins, including histone deacetylase [3,4,5,6,7,8,9,10,11,12,13,14,15,16,17,18,19,20,21,22,23,24,25,26,27,28,29,30,31,32,33,34,35,36,37,38].

Despite the above understanding, knowledge is still limited with respect to the exploration of the distinctive FFAs in ruminants such as cows that may show anticancer potential in the future. Herein, we report a suitable approach to extract and fractionate urine-derived metabolite fractions in sterile and cell-culture-compatible settings to investigate these aspects. Furthermore, we report an in-house approach assisted by vertical tube gel electrophoresis (VTGE) to identify specific cow-urine-derived metabolites in the intracellular compartment of treated cancer cells.

In this paper, we report a sound and novel approach to address and identify FFAs derived from cow urine DMSO fraction (CUDF) within the intracellular compartment of breast cancer cells treated with CUDF. Furthermore, these FFAs were studied for their molecular targets, including HDAC in cancer cells, using molecular docking and molecular dynamics (MD) simulations.

## 2. Materials and Methods

### 2.1. Materials

Cell culture reagents were purchased from Invitrogen India Pvt. Ltd. Bangalore, India and Himedia Laboratories Pvt. Ltd. Mumbai, India. MCF-7 and ZR-75-1 breast cancer cells were procured from the National Centre of Cell Science (NCCS), Pune, India. DMSO, pBR322 DNA, agarose, acrylamide, and other chemicals of analytical grade were obtained from Himedia Laboratories Pvt. Ltd. Mumbai, India. and Merck India Pvt. Ltd. Mumbai, India.

### 2.2. Preparation of CUDF

Fresh urine samples of cows were collected in a sterile falcon tube. Next, 20 mL of urine was oven-dried in a covered glass Petri dish to obtain solid materials. Then, solid materials of dried urine were reconstituted and redissolved in 1 mL DMSO solvent. Furthermore, 1 mL of DMSO containing urine components was transferred to a 2 mL Eppendorf tube for efficient extraction by employing intermittent vortexing for one h. At the end of vortexing, DMSO extract of cow urine was submitted to centrifugation twice at 12,000× *g* for 30 min to eliminate insoluble materials and particulates. The clear supernatant of DMSO extract was filtered using a sterile 0.45 micron syringe under aseptic conditions to obtain clear and sterile cow urine DMSO fraction (CUDF) in a sterile Eppendorf tube. Next, prepared CUDF was subjected to weighing by volume gravimetric analysis to calculate the final weight per volume. Finally, a sterile stroke concentration of CUDF at 10 mg/mL (weight per volume) was prepared for various cell-based assays of breast cancer cells.

### 2.3. Cell Line Maintenance and Preparation for Drug Treatment

MCF-7 and ZR-75-1 breast cancer cells were cultured and maintained in Dulbecco’s Modified Eagle Medium (DMEM) with high glucose supplemented with 10% heat-inactivated FBS/penicillin (100 units/mL)/streptomycin (100 µg/mL) at 37 °C in a humidified 5% CO_2_ incubator. Furthermore, 60–70% of confluent MCF-7 cells were harvested and subjected to drug treatment.

### 2.4. Trypan Blue Dye Exclusion Assay

The MCF-7 and ZR-75-1 breast cancer cells were cultured at 60–70% confluence. Cells grown overnight were plated into six-well plates at 150,000 cells per well. After 16–18 h of overnight growth, a complete DMEM medium with DMSO (10 µL) solvent control and CUDF at a final concentration 15 µg/mL, 25 µg/mL, and 50 µg/mL were added in triplicate wells of a six-well plate. Cells were allowed to incubate for 72 h; cells were observed, and their morphology was studied with the help of a routine microscopy technique. Then, cancer cells were harvested and collected using a routine trypsinization procedure. A routine trypan blue dye exclusion assay was performed to determine the number of total, viable, and dead cells.

### 2.5. MTT Cell Cytotoxicity Assay

MCF-7 and ZR-75-1 breast cancer cells were plated into a 96-well, flat-bottom plate at 10,000 cells per well. A complete volume of 200 µL of DMEM was added to each well. Breast cancer cells were allowed to grow for the next 16–18 h. Next, breast cancer cells were treated with CUDF (1 µL) at a final concentration (50 µg/mL) and DMSO control (1 µL) in triplicate. Breast cancer cells were incubated for 72 h at 37 °C. The rest of the procedure was adopted as per the routine protocol [39,40]. The absorbance was recorded using an ELISA reader (Thermo Fisher Scientific, Waltham, WA, USA).

### 2.6. Dual AO/EB Fluorescent Staining

MCF-7 and ZR-75-1 breast cancer cells were grown and treated as previously described for the trypan blue dye exclusion assay. At the end of treatment, cells were harvested with the help of trypsin and the cell suspension (25 µL = 500 cells), which was transferred to a glass slide. Then, dual fluorescent staining solution (1 µL) containing 50 µg/mL AO (Acridine orange) and 50 µg/mL EB (Ethidium bromide) (AO/EB, Sigma, St. Louis, MO) was added to each suspension and covered with a coverslip. The morphology of the apoptotic cells was examined, and 500 cells were counted within 20 min using a fluorescent microscope (Olympus Medical Systems India Private Limited, Gurgaon, India).

### 2.7. Flow-Cytometry-Based Apoptosis Assay

MCF-7 breast cancer cells were cultured in duplicate into six-well plates at a seeding density of 1.5 × 10^4^ cells per well. After 16–18 h, breast cancer cells were treated with CUDF (50 µg per ml) for 72 h and combined with 2 mL complete DMEM medium. Post treatment, breast cancer cells were pelleted and resuspended in 1 mL of cold PBS buffer. Next, annexin V binding buffer was added to the pellet obtained after centrifugation of MCF-7 cancer cells. The rest of the steps were followed as per the manufacturer’s instructions for the use of the annexin V/FITC apoptosis detection kit (ThermoFisher). Then, live and dead cells were recorded with a BD FACSJazz cytometer, and 10,000 events were measured for each sample [39,40].

### 2.8. Effects on Human Gingival Mesenchymal Stem Cells (hGMSCs)

GMSCs were cultured into six-well plates at a plating density of 15 × 10^4^ cells per well. After 16–18 h of plating, GMSCs were treated with CUDF and DMSO and incubated for 72 h, as previously described, in the presence of complete DMEM. Next, trypan blue dye exclusion and apoptosis assays were performed as described above to estimate the viability and cell death of GMSCs [39,40]

### 2.9. Preparation and Purification of Intracellular Metabolites by VTGE

To identify the intracellular CUDF-derived FFAs, breast cancer cells were suspended in hypotonic buffer (10 mM KCl, 10 mM NaCl, 20 mM Tris, pH 7.4) [39,40]. Three washing steps with PBS were performed to remove the traces of external medium components. Both viable and dead cells were accounted for while preparing whole intracellular lysates. Furthermore, 250 µL of whole-cell lysate was diluted to 750 µL by adding hypotonic buffer. Then, 250 µL of whole-cell lysate was mixed with 4X loading buffer (0.5 M Tris, pH 6.8, and Glycerol). Next, whole-cell lysate, along with loading buffer, was loaded in a vertical tube gel electrophoresis (VTGE) purification system (Appendix A) with a matrix of 15% acrylamide gel (acrylamide: bisacrylamide, 30:1). The fractionated intracellular metabolites were collected in 5× running buffer (96 mM glycine, pH 8.3). The detailed procedure was adopted from a previously published in-house VTGE-assisted metabolite purification method [39,40]. Furthermore, LC-HRMS analysis of intracellular metabolites was performed with an Agilent TOF/Q-TOF Mass spectrometer station Dual AJS ESI ion source. During LC separation, RPC18 Hypersil GOLD C18 100 × 2.1 mm − 3 µm and a mobile phase of 100% water (0.1% FA in water) and 100% acetonitrile (90% ACN + 10% H_2_O + 0.1% FA) were used in proportions of 95% and 5%, respectively [40]. Mass spectrometry was performed in positive mode and analyzed as per the procedure adopted from the previously reported methodology [40].

### 2.10. Molecular Docking

Potential intracellular FFAs, including tetracosanedioic acid (PubChem CID: 2724554), 13z-docosenoic acid (PubChem CID: 8216), nervonic acid (PubChem CID: 5281120), 3-hydroxy-tridecanoic acid (PubChem CID: 5312749), 8-hydroxy caprylic (PubChem CID: 69820), and trichostatin A (PubChem CID: 444732) as a positive control, were retrieved from the PubChem database (Sdf format, https://pubchem.ncbi.nlm.nih.gov, accessed on 30 June 2021); the database was used to download the structures of ligands in SDF format. Before performing molecular docking, all ligands were energy-minimized to obtain stable conformation using Avogadro software [41] with the steepest descent method and an MMFF94s force field. In this study, we considered the histone deacetylase (HDAC) receptor (PDB ID-1C3R), which was downloaded from the PDB database (https://www.rcsb.org, accessed on 30 June 2021) [42]. The preparation of proteins and molecular docking process by AutoDock Vina Software was adopted from a standardized protocol [42,43].

AutoDock Vina includes a feature to automatically calculate grid maps [44]. Confirmation of the binding position of potential FFAs into the cavity of the receptor and calculation of binding parameters such as bond distance were performed with Maestro.

### 2.11. Molecular Dynamics (MD) Simulations

The 10 ns molecular dynamics (MD) simulations for the complexes of tetracosanedioic acid (PubChem CID2724554) and the positive control, trichostatin A (PubChem CID: 444732) with histone deacetylase, were performed with the help of Desmond software to confirm the binding stability and strength of the complex [45]. Desmond has inbuilt functions to add pressure, system volume, and temperature, as well as many functionalities to accomplish protein–ligand binding. The first complex is a histone deacetylase with tetracosanedioic acid. The second complex is a well-known inhibitor (FDA-approved), trichostatin A, which was used as a positive control in complex with histone deacetylase. The first complex was immersed in 9071, and the second was 7350 TIP3P explicit water molecules within a cubic box with 10 Å spacing using periodic boundary conditions. An MD simulation study was carried out with a run of 10 ns at a temperature of 300 K considering specific parameters such as the integrator as MD. The conformational changes upon binding of FFAs with histone deacetylase were recorded in 1000 trajectories of frames generated during the 10 ns MD simulation. Root mean square deviation (RMSD) was calculated to reveal the binding stability of tetracosanedioic acid and trichostatin A with the HDAC complex. The steepest descent method was used to energetically minimize the complex system with the OPLS-2005 force field. Then, 10 ns time scale MD simulations were performed for each complex at a constant NPT (N = number of atoms; P = pressure; and T = temperature) ensemble. Throughout the equilibrations, systems were coupled with the Martyna–Tobias–Klein barostat method to control pressure at 1 atm. The temperature was regulated using the velocity-rescaling Nose–Hoover chain thermostat method at 300 K. The M-SHAKE algorithm was used to constrain the bond length of hydrogen atoms. The cutoff for short-range electrostatics and van der Waals interactions were maintained at 1 nm. Long-range Coulomb electrostatic interactions were calibrated through particle mesh Ewald (PME) summation. The leap-frog algorithm was used to compute the equation of motion with a time step of 2 fs [41]. The conformational changes in histone deacetylase at C-α backbone atoms upon binding of tetracosanedioic acid were compared with the initial conformations of (crystal structure) a histone deacetylase (PDB ID: 1C3R) in terms of root mean square deviation (RMSD).

## 3. Results

### 3.1. Effects of CUDF on Cell Viability

Recently, FFAs have been highly appreciated for their anticancer role and have been reported to induce apoptotic cell death in various cancer cell types, including breast cancer cells [5,6,7,8,9,10]. Based on the limited idea of the potential of FFAs and other organic acids as anticancer compounds, FFA-enriched CUDF compositions were treated in MCF-7 and ZR-75-1 breast cancer cells in a dose-dependent manner. A simple and commonly used trypan blue dye exclusion assay was performed to estimate the effects of CUDF in reducing total cells and loss of cell viability in these breast cancer cells. Photomicrographs at 100× of MCF-7 cancer cells treated by CUDF demonstrated total cell and cell viability reduction in the DMSO control (Figure 1A). Total and viable breast cancer cells were estimated by hemocytometer counting. CUDF treatment resulted in a reduction in cell viability of up to 41.63% relative to the DMSO-control-treated MCF-7 cells at a 50 µg/mL concentration (Figure 1B) (*p* ≤ 0.001). It is important to note that CUDF decreased the total cells to 51.36% (*p* ≤ 0.001) compared to DMSO. This observation suggests the possibility of the role of CUDF in the proliferative arrest of MCF-7 breast cancer cells.

The loss of cell viability was also determined by an MTT assay. Data indicated that CUDF induced loss of MCF-7 cell viability up to 62.69% by normalizing to the DMSO control (Figure 2A,B) (** *p* ≤ 0.01). Reduction in cell viability as a percentage is an indirect measurement of cancer cell toxicity based on the production of formazan crystals due to the activity of a mitochondrial enzyme. Therefore, the assessments of cell viability of CUDF-treated MCF-7 cells by trypan blue dye exclusion and MTT assays differ and suggest possible means of observation of loss of viability.

Next, in order to estimate the nature of cell death in MCF-7 cancer cells, a fluorescence microscopy-based acridine orange (AO)/ethidium bromide (EO) dual staining assay was performed. In the essay, a study of nuclear morphology and intracellular staining by AO/EB was performed. Fluorescent microscopy photographs of AO/EO dual-stained, DMSO-treated MCF-7 cells were mainly stained with uniform green fluorescence and showed the negligible presence of apoptotic cells (left panel, Figure 3A). In the case of MCF-7 cancer cells treated with CUDF, noticeable morphological changes and a clear presence of apoptotic MCF-7 cells with chromatin condensation can be observed in the right panel of Figure 3A. Here, MCF-7 cells stained with green represent viable cells, yellow represents early apoptotic cells, reddish colors show necrosis, and dark-orange-stained cells represent late apoptotic cells. These observed viable, necrotic, and apoptotic cells were scored by counting and represented as the percentage of total cells normalized to the DMSO control (Figure 3B). Data suggest that CUDF produced a significant loss of cell viability (** *p* ≤ 0.01), as well as the presence of apoptosis (** *p* ≤ 0.01) in MCF-7 cancer cells.

To estimate the apoptosis-inducing potential, DMSO- and CUDF-treated MCF-7 cancer cells were stained with annexin V and PI and measured using a flow cytometer. The scatter plot of the distribution of viable cells, non-viable cells, early apoptotic cells, and late apoptotic cells in DMSO, and CUDF-treated MCF-7 breast cancer cells is presented in Figure 4A. A reduction in cell viability of MCF-7 cancer cells from 79.64% (DMSO control) to 60% is observed due to CUDF treatment (Figure 4B). Based on the cell counts during the annexin/PI staining assay, the percentage of apoptotic MCF-7 cells was calculated and normalized to the DMSO control, appreciably showing the presence of 58.27% apoptotic cells in CUDF-treated MCF-7 cells relative to the DMSO control (Figure 4C).

Besides the effects of CUDF on MCF-7 breast cancer cells, other breast cancer ZR-75-1 cells were treated with CUDF in a dose (15 µg/mL, 25 µg/mL, and 50 µg/mL)-dependent manner. Data supported an apparent reduction in the total viable cells up to 34.08% (** *p* ≤ 0.01) (Appendix A). Here, observations supported the distinct response of CUDF in ZR-75-1 breast cancer cells in terms of a reduction in the total number cells and loss of the viability of cells. In another way, CUDF possibly showed a loss of cell viability independent of a reduction in total cells. However, in the case of MCF-7 cells, CUDF showed a similar trend of reduction in total cells and loss of viability in a dose-dependent manner.

Additionally, analysis of AO/EO dual-stained ZR-75-1 breast cancer cells indicated a clear presence of apoptotic cell death up to 35.15% (** *p* ≤ 0.01) relative to the DMSO control and, to a certain extent, necrotic cells (Appendix A). Fluorescence microscopy clearly illustrated a significant number of apoptotic and necrotic cells in the case of CUDF-treated ZR-75-1 breast cancer cells. The presence of apoptotic cells in these ZR-75-1 breast cancer cells was estimated using a PI/annexin staining assay. These observations also indicated the presence of up to 20% apoptotic cells normalized to DMSO in CUDF-treated ZR-75-1 cells (Appendix A–C). CUDF appears to induce significant cell death in ZR-75-1 breast cancer cells; this observation aligns with the role of CUDF in MCF-7 breast cancer cells. In summary, the IC_50_ value of FFA-enriched CUDF was found to be similar: 39.64 and 38.88 µg per ml for MCF-7 and ZR-75-1 breast cancer cells, respectively. However, molecular mechanisms of FFA-enriched CUDF as a potential agent for proliferative arrest and cell death in MCF-7 and ZR-75-1 cells could be distinct and warranted in the future with detailed investigations in cell-based, preclinical, and clinical models.

### 3.2. Effects of CUDF on Normal Gingival Mesenchymal Stem Cells (GMSCs)

Besides observing CUDF-mediated loss of cell viability and the induction of apoptotic cell death in MCF-7 cancer cells, gingival mesenchymal stem cells (GMSC) were selected as normal counterpart cells to understand the effects of CUDF on non-cancer cells. GMSCs were treated with CUDF and evaluated for cell viability using a routine trypan blue dye exclusion assay. Photomicrographs did not show any significant changes in the cellular morphology or loss of viability in CUDF-treated GMSCs relative to the DMSO control (Figure 5A). Data indicated no significant reduction in the cell viability of GMSCs treated with CUDF relative to the DMSO control (Figure 5B). Furthermore, no discernible presence of apoptotic cells was noticed with the help of annexin V and PI staining of GMSCs treated with CUDF (Figure 5C). In summary, CUDF did not display any observable toxicity or induce cell death in GMSCs.

### 3.3. Intracellular Metabolite Profiling

Based on the antiproliferative and proapoptotic effects of CUDF enriched with FFAs on MCF-7 cells, logical questions arose to pinpoint and identify specific chemicals such as organic acids and FFAs that entered the intracellular levels of treated MCF-7 cells. Obvious questions included whether there were any distinctions and specificities compared to previously reported papers on cow urine and other similar extract-based anticancer compositions. In essence, intracellular metabolite profiling of cancer cells treated by CUDF and other known anticancer drugs is highly limited due to the incompatibility of approaches. To address the need to identify intracellular chemicals derived from CUDF, we employed a novel in-house methodology assisted by VTGE and LC-HRMS analysis.

Based on observations that suggested appreciable antisurvival and pro-cell-death effects of FFA-enriched CUDF on MCF-7 cancer cells, a pertinent question was raised about the components of CUDF that entered into the intracellular compartment of MCF-7 cancer cells. An in-house developed VTGE-assisted profiling method for intracellular metabolites, including FFAs, is presented here.

A representative LC-HRMS total ion chromatogram (TIC) is presented for intracellular metabolites of DMSO- and CUDF-treated MCF-7 cells in Figure 6A,B, respectively. The analysis of TIC suggested that specific FFAs such as tetracosanedioic acid (RT-14.871), 13Z-docosenoic acid (erucic acid, RT-10.896), nervonic acid (RT-12.835), 3-hydroxy-tridecanoic acid (RT-14.322), and 8-hydroxycapric acid (RT-0.89) were detected in the intracellular compartment of CUDF-treated MCF-7 cells relative to the DMSO control. The characteristic MS fragmentation ions of these FFAs are noted as tetracosanedioic acid (*m*/*z* 282.276, 398.3465, 399.3537, and 419.0823), 13Z-docosenoic acid (*m*/*z* 240.2298, 343.2923, and 419.0830), nervonic acid (*m*/*z* 244.2245, 366.3439, 371.3226, and 419.0823), 3-hydroxy-tridecanoic acid (*m*/*z* 235.1664, 284.3279, and 419.0824), and 8-hydroxycapric acid (*m*/*z* 144.0609, 193.1158, and 215.0974), and these ion spectra matched with available databases (Table 1 and Appendix A, respectively). According to the MS fragmentation ion spectra, tetracosanedioic acid, 13Z-docosenoic acid, and nervonic acid share a backbone structure, and a distinction in structure is noticed at the level of the degree of unsaturation and oxidation.

### 3.4. Molecular Docking

Based on the favorable evidence of the intracellular presence of selected FFAs in MCF-7 cells treated by CUDF, we raised questions on the relevance of these FFAs within the intracellular compartment of MCF-7 cancer cells concerning the observed apoptotic cell death. We also attempted to link the molecular interactions of these selected FFAs with key oncoproteins, including HDAC. In this study, we performed oncoprotein–FFA molecular docking to screen the interactive potential of these FFAs. Several potential oncoproteins targets were selected for initial evaluation and screening as potential targets of FFAs (Appendix A). Based on the initial screening, these FFAs were further evaluated as potential histone deacetylase inhibitors (PDB ID: 1C3R). Furthermore, details of site-specific molecular docking data on histone deacetylase (PDB ID: 1C3R) with FFAs indicated appreciable binding affinities ranging from −6.0 to −7.5 (kcal/mol) (Table 2). Among selected intracellular FFAs, tetracosanedioic acid displayed the highest binding affinity value (−7.5 kcal/mol) against HDAC, and its binding positions were within the inhibitory domain (Figure 7A,C). We also performed molecular docking for trichostatin A, a known inhibitor of HDACs, for comparison with tetracosanedioic acid and predicted a resemblance of the binding affinity (−7.9 kcal/mol) (Table 2). DSV3 data for tetracosanedioic acid, 13-docosenic acid, and nervonic acid suggested similar binding positions that included key inhibitory-domain amino acid residues such as Tyr297, Leu265, Phe200, Phe198, Gly140, His132, Tyr91, Pro22, and His21. These key inhibitory binding residues are also bound by a known inhibitor of HDACs, trichostatin A (Table 2, Figure 7B,D). The binding affinities and nature of molecular interactions of selected FFAs tetracosanedioic acid, 13-docosenic acid, and nervonic acid coincided with trichostatin A and are appreciably comparable. An important note is that two other FFAs, i.e., 3-hydroxy-tridecanoic acid and 3-hydroxycarpic acid, did exhibit affinity to bind within the inhibitory sites. These observations suggest that the chemical nature of hydroxylated FFAs is distinct; therefore, the specificity of tetracosanedioic acid, 13-docosenic acid, and nervonic acid against HDACs is precise and logical.

### 3.5. Molecular Dynamics (MD) Simulations

Based on the initial screening and visualization of the HDAC binding domain with tetracosanedioic acid and TSA, MD simulations of tetracosanedioic acid and HDAC interactions were performed, the results of which are presented as an RMSD graph (Figure 8A). These data define the conformational changes upon binding of tetracosanedioic acid with HDACs, which were recorded using the 1000 trajectories frames generated during the 10 ns MD simulations. The slightest changes in the RMSD value of HDAC indicate better protein stability upon binding with the ligand tetracosanedioic acid during the 10 ns MD simulation. Analysis of the RMSD plot suggested complex stability between tetracosanedioic acid and HDACs, and deviation is noted in the range of 0.8 to 1.8 Angstrom. Interestingly, MD simulation data on the known inhibitor, TSA with HDAC, is well aligned with the RMSD plot of the HDAC–tetracosanedioic acid complex, which showed similar amino acid residues at inhibitory binding pocket, as well as of RMSD value deviation in the range of 1–2 Angstrom (Figure 8B).

Furthermore, the protein ligand contact map of tetracosanedioic acid (Figure 9A), which coincided with a known inhibitor TSA (Figure 9B) in complex with HDAC, reveals common inhibitory domain residues, such as Gly129, His131, His132, Asp168, Phe198, and Tyr297. Molecular interaction fractions for interacting amino acid residues for both tetracosanedioic acid and TSA displayed similar values, ranging from 0.1 to 0.8 A.U. Analysis of root mean square fluctuation (RMSF) of amino acid residues of HDAC revealed an interesting observation that both tetracosanedioic acid and TSA did not induce any notable fluctuations within key inhibitory domain residues, such as Gly129, His131, His132, Asp168, Phe198, Phe200, Asp258, and Tyr297 (Figure 10). The RMSF plot suggests that minimal fluctuation in the inhibitory domain was observed upon binding tetracosanedioic acid and TSA compared to the apo form of HDAC. A proposed model of the role of FFAs as antiproliferative and proapoptotic molecules with an inhibitory potential upon HDACs is depicted in Figure 11.

### 3.6. vNN Web Server ADMET Predictions

A relevant question was raised regarding the toxicity profile of selected FFAs such as carcinogenicity and its affinity with P-gp. To address these concerns, vNN-ADMET was used and predicted that tetracosanedioic acid is not a suitable substrate of P-gp, predicting no carcinogenicity. Tetracosanedioic acid was predicted as superior in terms of non-toxicity, non-carcinogenicity, and the highest recommended maximum therapeutic dose (MRTD) of 15,601 mg/day (Appendix A). Conversely, TSA, a known inhibitor of HDAC with MRTD at 150 mg/day, was predicted as a substrate of P-gp with potential carcinogenicity (Appendix A).

## 4. Discussion

Genotoxic drugs such as cisplatin and doxorubicin are known for discernible drug resistance. There are efforts to look for new classes of safer drugs, including FFAs, derived FFAs, and mimetics of FFAs. Cancer cells employ FFAs and their products from fatty acid metabolisms for their growth and proliferation by creating components of the cell membrane, a source of energy, and secondary messengers [3]. FFAs are diverse, and cancer cells display heterogeneity in terms of preferences for certain types of FFAs for their metabolic needs over cell-death-inducing FFAs. The primary sources of FFAs are dietary and microbial fermentations by gut-associated microbiotas and cellular metabolisms. Therefore, a better understanding of the metabolic pathways of FFAs and their profile is a potential avenue for anticancer approaches.

Intracellular requirements of fatty acid profiles in breast cancer cells differ from the routinely available exogenous fatty acids in the form of dietary lipids, fatty acids, and microbial fermentation [30,31]. In the case of exogenous FFAs as anticancer drugs, the uptake of free fatty acids is facilitated by passive diffusion and, to a certain extent, by the available CD36 expression [30,31]. After entry, these fatty acids are seen as molecular modulators of various signaling pathways beyond the energy needs of breast cancer cells. Therefore, exogenous uses of fatty acids as anticancer drugs are warranted in terms of their abilities to modulate specific protumor metabolic signaling pathways.

To test the above question, it was necessary to find the sources of compositions of FFAs (COFFA) as antiproliferatives and inducers of apoptotic cell death. Furthermore, the proportions of antitumor FFAs and protumor FFAs are associated with dietary intake and gut-microbiome-based fermentations of FFAs. This proposition allows for the presence of metabolically converted new sets of FFAs that are implicated in disease conditions, including cancer [17]. Based on this premise, a relevant question is raised with respect to the selection FFAs as DMSO fractionated cow urine metabolites. Cow urine and other biological fluids may represent a good source of dietary-derived chemicals, gut microbial fermentation products, and tissue-based metabolic products.

Additionally, cows and other ruminants have been reported for their rare occurrence of breast tumors compared to humans. However, xeno-tumor heterogeneity has not been explored, which can hint at the potential role of dietary-derived metabolic products in cows and other ruminants. This information prompted us to consider the cow urine metabolites fractionated in DMSO as compositions of metabolites, specifically as the composition of FFAs (COFFAs), because FFAs such as 13Z-docosenoic acid, nervonic acid, and tetracosanedioic acid are exclusively soluble in DMSO.

In this study, CUDF was prepared and fractionated in DMSO solvent using a previously reported standardized method. CUDF containing enriched FFAs and other organic acids (data not shown) was prepared. CUDF enriched with FFAs was filtered by a 0.4 -micron sterile syringe filter to make it compatible with cell-based and metabolite profiling assays. DMSO is the best solvent for long-chain FFAs compared to aqueous solvents such as water. Existing papers on cow-urine-based formulations and extractions adopted an aqueous-based solvent system that might have excluded the presence of long-chain FFAs, as observed in our DMSO-based FFA-enriched CUDF. The novel approach proposed in this paper facilitated the detection of long-chain FFAs in the intracellular compartment of treated breast cancer cells.

The role of 13Z-docosenoic acid (erucic acid) has been documented in the context of anti-inflammatory and anticancer effects. On the other hand, nervonic acid and tetracosanedioic acid, which are derived from erucic acid, are less reported for their biological activities, including induction of cancer cell death. 13Z-docosenoic acid is mainly derived from dietary sources. Nervonic and tetracosanedioic acid are related to biological sources such as tissue and gut microbial fermentation of dietary FFAs. However, there have been no reports on the role of 13Z-docosenoic acid, nervonic acid, and tetracosanedioic acid as dietary and gut microbial compositions to induce cell death in target cancer cells, including breast cancer cells.

Data reported in this paper suggested that CUDF treatment of MCF-7 and ZR-75-1 breast cancer cells leads to substantial loss of cell viability and apoptotic cell death. It is essential to highlight that the anticancer effects of FFA-enriched CUDF on two different types of breast cancer cells, i.e., MCF-7 and ZR-75-1, are clear and significant. At the same time, observations of viability and cell death in these two types of breast cancer cells are distinct. These data can be explained and linked with their caspase 3-deficient MCF-7 cells relative to caspase 3-active ZR-51-1 breast cancer cells [45,46,47]. Despite similar WT p53 profiles in MCF-7 and ZR-75-1 breast cancer cells, the status of caspase 3 is distinct. Therefore, a careful interpretation of the effects of CUDF as antiproliferative or proapoptotic agents on these distinct profiles of breast cancer cells needs to be drawn. In the future, preclinical and clinical evaluation of COFFAs should consider these genetic profiles, including caspase 3 and p53 status in breast cancer cells.

Data suggested that CUDF may contribute to more apoptotic cell death in ZR-75-1 cells than in MCF-7 cells. Furthermore, identification of FFAs such as tetracosanedioic acid, 13-docosenoic acid, and nervonic acid in the intracellular compartment of CUDF-treated MCF-7 cancer cells suggests that these FFAs reported from dietary and microbial fermentation may contribute to the observed loss of viability and apoptotic cell death. Our findings align with current views on the modulatory effects of selected FFAs on cancer cells.

Previous findings support the association between the heterogeneous nature of FFAs in the diet and dietary supplements, as well as possible antitumor and protumor implications [7,10]. Among various FFAs, erucic and stearic acids are suggested for antitumor effects in nature. Conversely, linoleic acids, oleic acid, and palmitic acid are indicated for protumor effects [10,11,12,16,17,18,19,20,21,22,40,41,43,44,45]. Studies suggest that distinctive positions of double bonds and additional functional groups such as hydroxyl groups and carboxyl groups in unsaturated fatty acids obstruct conversion into prostaglandins, which are well-known agents in inflammation and cancer proliferation [40,41,43,44,45].

Nutritional and epidemiological evidence converge, suggesting that the growth and progression of cancer cells are linked with the chemical profile of saturated and unsaturated FFAs. In the literature, selected FFAs are known for their anticancer action and as inducers of cancer cell death [16,17,18,19,20,21,22]. Conversely, some FFAs are also reported for their carcinogenic potential. Taken together, FFAs and their modified metabolic products can be explored for their antiproliferative and apoptotic effects in cancer cells. Several saturated and unsaturated fatty acids are converted into carboxylated and hydroxylated forms in bacterial, plant, and mammalian systems with the help of their inherent metabolic enzymes [18,37,38,39,40,41]. The data that substantiate the action of these derived forms of fatty acids as carboxylated and hydroxylated in cancer cells are minimal [21,41]. Furthermore, profiling of saturated and unsaturated FFAs in the intracellular compartment of breast cancer cells treated with exogenous FFAs is limited by the lack of appropriate metabolite profiling approaches.

Interestingly, the role of these selected long-chain saturated dicarboxylic acids, such as tetracosanedioic acid (22:0), has not been explored as an agent to induce cancer cell death. A possibility of enzymatic conversion of erucic acid and nervonic acid by microbial and animal cells has been proposed, but with few supporting studies. Therefore, the combined presence of long-chain FFAs, including erucic acid, nervonic acid, and tetracosanedioic acid, is related to the fatty acid metabolism of cows and their gut microbial system since these FFAs are derived from CUDF.

Recently, selected FFAs such as oleic acid, DHA, and other long-chain saturated and unsaturated FFAs have been reported for their antiproliferative and proapoptotic effects in cancer cells. These selected FFAs are mainly inhibitors of pro-proliferation proteins such as CDK–cyclin and antiapoptotic proteins. Therefore, the relevance of 13Z-docosenoic acid, nervonic acid, and tetracosanedioic acid as proapoptotic agents that can inhibit potential protumor proteins such as HDACs is logical and required improved understanding.

HDACs are known for their crucial role in remodeling chromatin and non-chromatin proteins that impact various biological processes, including their contribution to growth, proliferation, and suppression of apoptotic cell death [25]. Epigenetic modulations by HDACs influence the transcription state of various key genes, including oncoproteins and tumor suppressors. Therefore, inhibitors of HDACs are emphasized as new classes of anticancer drugs other than genotoxic drugs. These inhibitors of HDACs induce the arrest of cellular growth and the promotion of cellular death in cancer cells. Limited studies have reported a reduction in the activity of HDAC by short-chain FFAs, long-chain FFAs, and other classes of small molecules concerning immune modulation and cancer cell survival [27,28].

There is substantial evidence of the induction of apoptotic cell death in cancer cells treated with HDAC inhibitors such as butyrate, suberoylanilide hydroxamic acid (SAHA), and trichostatin A (TSA) [23,24,25,26,27,28,40,41]. A recent study indicated that saturated fatty acids such as pentadecanoic acid and others inhibited HDAC6, resulting in the inhibition of MCF-7 breast and A549 lung cancer cells [23,24,25,26,27,28]. Additionally, molecular docking suggests that an increased aliphatic carbon chain may facilitate the enhanced binding of HDAC6 residues. Information on the intracellular presence of selected FFAs in breast cancer cells treated with FFA compositions and their interactions with key protumor proteins such as HDACs is inadequate.

Our finding is incremental, as the new classes of inhibitors of HDACs in the form of selected FFAs such as erucic acid, nervonic acid, and tetracosanedioic acid can induce cell death in MCF-7 cancer cells. In this study, molecular docking and MD simulations were conducted to compare the HDAC inhibitory potential of tetracosanedioic acid and TSA. Data convincingly indicated that the binding affinity, specificity, and stability of molecular interactions are almost identical. Tetracosanedioic acid is a dicarboxylic saturated fatty acid and a known inhibitor of HDAC. TSA is a monocarboxylic acid. The chemical structure during the modeling of these two compounds is similar, which can account for their comparable HDAC inhibitory activity. In this paper, we have highlighted the presence of saturated and unsaturated long-chain fatty acids in the intracellular compartment of breast cancer cells. These long-chain FFAs are suggested to work as excellent candidates to inhibit HDACs with the help of molecular docking and MD simulations. Taken together, cell-based and MD simulations suggested tetracosanedioic acid and other long-chain FFAs as inhibitors of HDAC, which may account for intracellular effects leading to antiproliferative and proapoptotic effects in breast cancer cells.

Hydroxyl FFAs are a unique class of FFAs known for various biological activities, including antibacterial, antifungal, and anticancer activities [30,37,40,41,43,44,45]. Interestingly, hydroxylated FFAs derived from CUDF are not suggested as inhibitors of HDACs. Such observations indirectly show that hydroxylated FFAs such as microbially fermented FFAs may have other intracellular anticancer functions in CUDF-treated cancer cells. In the future, target proteins of hydroxylated FFAs in cancer cells can be explored in silico and in vitro.

Various forms of long-chain fatty acids obtained from dietary and natural sources have been explored for their anticancer and antibacterial effects by interfering with known molecular targets such as topoisomerase enzymes [48,49,50,51,52]. In the same line, we propose exploration of the effects of COFFAs with a specific inhibitory role against topoisomerase enzymes, including tetracosanedioic acid, 13Z-docosenoicacid (erucic acid), nervonic acid, 3-hydroxy-tetradecanoic acid, and 3-hydroxycapric acid, which are known as potential anticancer drug targets.

## 5. Future Perspectives

This paper provides a platform to discuss xeno-tumor heterogeneity, which is an interesting aspect in the cancer field to understand the contributing factors that create distinctive susceptibility and resistance across the species, such as humans versus ruminants such as cows.Ruminants such as cows and goats are known for rare occurrence of mammary cancers relative to humans, who show high susceptibility.There are various explanations for this difference, including evolution, genetic adaptations, and mutations. Nevertheless, the abundance of a set of FFAs derived from dietary and gut microbial fermentations, as well as their potential interference with cancer cell growth and proliferation, requires detailed investigations in the future to support the idea of xeno-tumor heterogeneity.Evaluation of FFA-enriched CUDF as antiproliferative and proapoptotic agents should be carefully interpreted based on the caspase 3 and p53 status in breast cancer cells.The inhibitory role of these COFFAs against HDAC is warranted at in vivo, preclinical, and clinical levels as anticancer pharmaceutical compositions.Our propositions with respect to the role of FFAs as inhibitors of HDAC and their potential role as a source of antiproliferative and proapoptotic agents should be viewed as a source of combinatorial therapies with specific consideration regarding the status of caspase 3 and p53 status in breast cancer cells.COFFAs proposed as anticancer compositions can usually cross the cell membrane in a passive diffusion process. However, the use of nanocarriers for better drug delivery of COFFAs is also proposed.

## 6. Conclusions

The dietary intake of FFAs are differs from one animal to another, including humans. Such FFAs, including omega-6 FFAs, omega-3 FFAs, and omega-9 FFAs, are converted in the gut system by human cells and gut bacteria. During the conversion of FFAs from dietary sources, additional beneficial FFAs are generated that are not well-known due to limitations of experimental approaches. In this study an in-house VTGE-assisted approach helped to identify FFAs, including tetracosanedioic acid, 13Z-Docosenoic acid, nervonic acid, tetracosanedioic acid, 3-hydroxy-tridecanoic acid, and 3-hydroxycapric acid, within the intracellular compartment of MCF-7 cancer cells treated with CUDF enriched with FFAs. This first report identifies FFAs such as tetracosanedioic acid, 13Z-docosenoic acid, nervonic acid, tetracosanedioic acid, 3-hydroxy-tridecanoic acid, and 3-hydroxycapric acid at an intracellular level of breast cancer cells showing proliferative arrest and induction of apoptotic cell death in MCF-7 and ZR-75-1 breast cancer cells. Molecular docking and MD simulations strongly suggested the affinity of tetracosanedioic acid, 13Z-docosenoic acid, and nervonic acid as inhibitors of HDAC. These selected FFAs, including tetracosanedioic acid, displayed comparable and matched inhibitory potential against HDAC, as reported previously for a known inhibitor, TSA. Altogether, these findings propose that selected FFAs, including tetracosanedioic acid from dietary sources and microbial fermentation, are feasible candidates in the form of COFFAs to target breast cancer cells.

## 7. Novelty and Impact Statements

This paper reports on intracellular free fatty acids (FFAs) derived from cow urine DMSO fraction (CUDF) during the treatment of MCF-7 breast cancer cells;This observation is the first report on the evaluation of intracellular FFAs in CUDF-treated MCF-7 cells by employing a novel in-house developed vertical tube gel electrophoresis (VTGE) system that helped in intracellular metabolite profiling;Furthermore, molecular docking and molecular dynamics (MD) simulations predicted the role of FFAs as inhibitors of HDACs that may potentially link the apoptotic effects in MCF-7 cells by FFA-enriched CUDF;This FFA fraction does not cause cell death in human gingival mesenchymal stem cells (hGMSCs), indicating its nontoxic effect on normal cells;This is the first report discriminating normal cells from cancer cells by employing FFAs derived from CUDF.

## Figures and Tables

**Figure 1 biomedicines-11-00889-f001:**
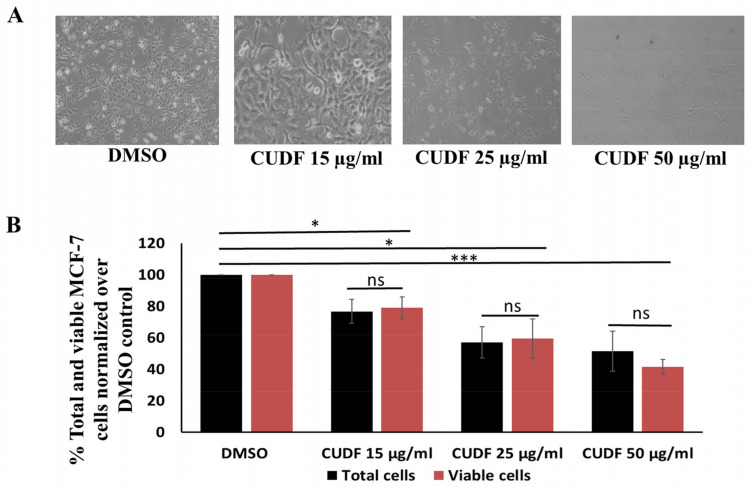
Cow urine DMSO fraction (CUDF) induces loss of cell viability in MCF-7 breast cancer cells. (**A**) MCF-7 cancer cells were treated with DMSO and CUDF (15 µg/mL, 25 µg/mL, and 50 µg/mL) for 72 hr. Routine microscopy was performed at 100× to observe the cell number and cellular morphology. (**B**) MCF-7 cancer cells were treated with DMSO and CUDF (15 µg/mL, 25 µg/mL, and 50 µg/mL) for 72 h. The percentage total and cell viability of MCF-7 cancer cells were estimated by trypan blue dye exclusion assay and normalized relative to the DMSO control. Data are represented as mean ± SD. Each experiment was conducted independently three times. The bar graph without an asterisk denotes no significant difference compared to the DMSO control. * Significantly different from the DMSO control at *p*-value ≤ 0.05. *** Significantly different from the DMSO control at *p*-value ≤ 0.001.

**Figure 2 biomedicines-11-00889-f002:**
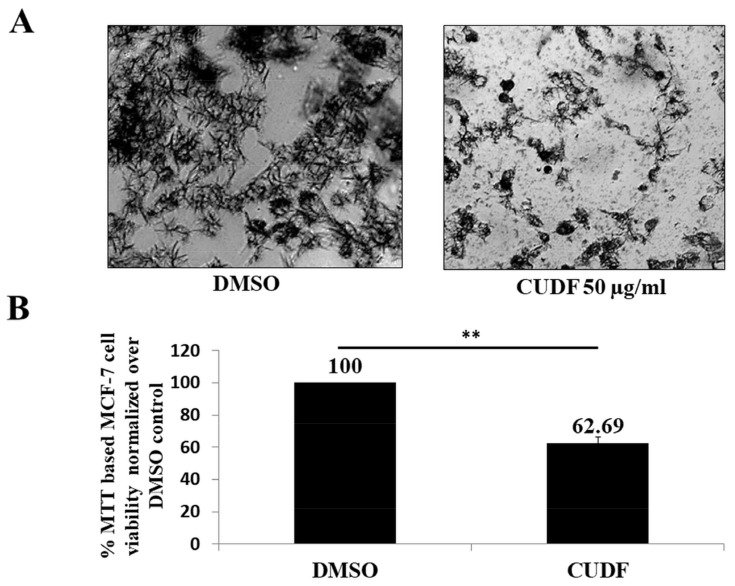
Cow urine DMSO fraction (CUDF) shows cytotoxicity to MCF-7 breast cancer cells. (**A**) MCF-7 cancer cells were treated with DMSO and CUDF (50 µg/mL) for 72 h. During the MTT assay, microscopic images of viable cells were captured, and ELISA-based measurements were performed. (**A**) Microscopic photographs captured at 100×. (**B**) MCF-7 cancer cells were treated with DMSO and CUDF (50 µg/mL) for 72 h. The value in terms of the percentage of cell viability normalized relative to the DMSO control was calculated based on the absorbance obtained in the MTT assay. ** Significantly different from the DMSO control at a *p*-value ≤ 0.01.

**Figure 3 biomedicines-11-00889-f003:**
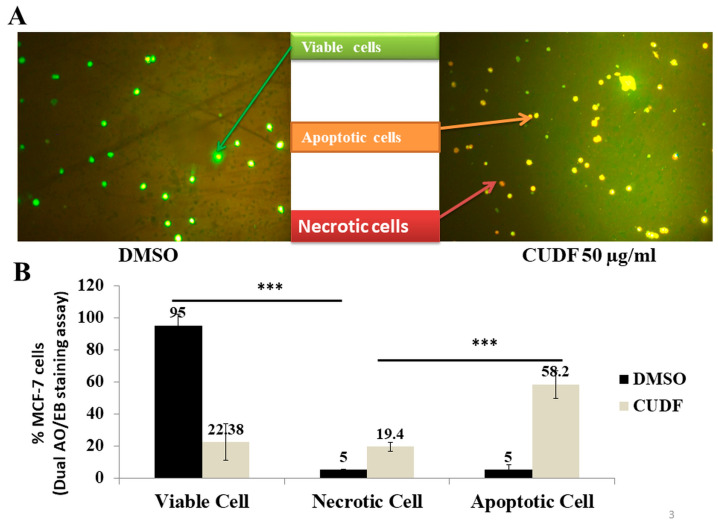
CUDF displays the presence of apoptotic cells in MCF-7 cancer cells by dual acridine orange (AO)/ethidium bromide (EB) staining. (**A**) MCF-7 cancer cells were treated with DMSO and CUDF (50 µg/mL) for 72 h. At the end of incubation, cells were harvested and processed for acridine orange (AO)/ethidium bromide (EB) staining as described in the Experimental Procedure section. Fluorescent microscopic images were captured at 40× magnification. (**B**) The percentages of viable cells, necrotic cells, and apoptotic cells among MCF-7 cancer cells were estimated by counting cells differentially stained with AO/EB. Green color: normal cells; yellow-green fluorescence by acridine orange (AO): early apoptotic cells; crescent or granular group of cells shows: late apoptotic cells; EB-stained orange fluorescence: necrotic cells. Data are represented as mean ± SD. Each experiment was conducted independently three times. The bar graph without an asterisk denotes no significant difference compared to the DMSO control. *** Significantly different from the DMSO control at a *p*-value ≤ 0.001.

**Figure 4 biomedicines-11-00889-f004:**
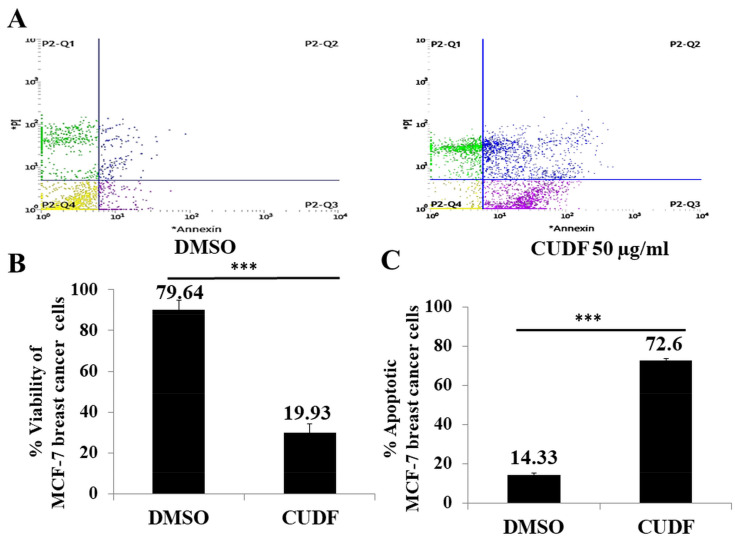
Estimation of CUDF-induced apoptosis in MCF-7 cancer cells. (**A**) MCF-7 cancer cells were treated with DMSO and CUDF (50 µg/mL) for 72 h. At the end of treatment, harvested MCF-7 cells were subjected to PI/annexin V staining and analyzed by a flow cytometer. The scatter plots of cells were stained with PI and Annexin V conjugated with FITC for analysis of apoptotic cells in MCF-7 cells. (**B**) MCF-7 cancer cells were treated with DMSO and CUDF (50 µg/mL) for 72 hr. At the end of treatment, harvested MCF-7 cells were subjected to PI/annexin V staining and analyzed with a flow cytometer. The percentage of cell viability was calculated based on the number of PI- and annexin-V-negative cells in quadrant 4 of the scatter plot divided by the total number of cells. (**C**) MCF-7 cancer cells were treated with DMSO and CUDF (50 µg/mL) for 72 h. At the end of treatment, harvested MCF-7 cells were subjected to PI/annexin V staining and analyzed with a flow cytometer. The percentage of apoptotic cells was calculated based on the number of cells in quadrants 3 and 4 stained with annexin V and PI divided by the total number of cells. Data are represented as mean ± SD. Each experiment was conducted independently three times. *** Significantly different from the DMSO control at a *p*-value ≤ 0.001.

**Figure 5 biomedicines-11-00889-f005:**
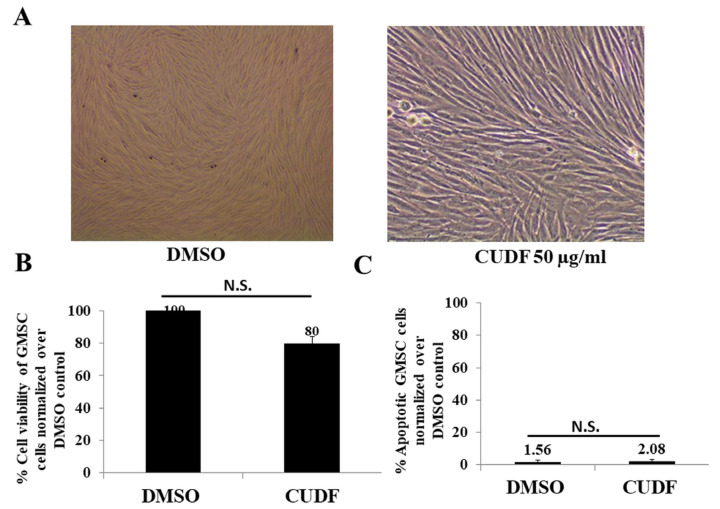
Effects of CUDF upon GMSCs in terms of cell viability and apoptotic cell death. (**A**) GMSCs were treated with DMSO and CUDF (50 µg/mL) for 72 hr. Photomicrographs were captured at 100× magnification at the end of incubation for 72 h. (**B**) GMSCs were treated with DMSO and CUDF (50 µg/mL) for 72 h. The percentage of cell viability was estimated by a trypan blue due exclusion assay. (**C**) GMSCs were treated with DMSO and CUDF (50 µg/mL) for 72 h. At the end of incubation, GMSCs were stained with PI and annexin V, and stained cells were measured by a flow cytometer. The percentage of apoptotic MCF-7 cells was estimated by counting the percentage of PI- and annexin-V stained MCF-7 cells. Data are represented as mean ± SD. Each experiment was conducted independently three times. The bar graph without an asterisk denotes no significant difference compared to the DMSO control.

**Figure 6 biomedicines-11-00889-f006:**
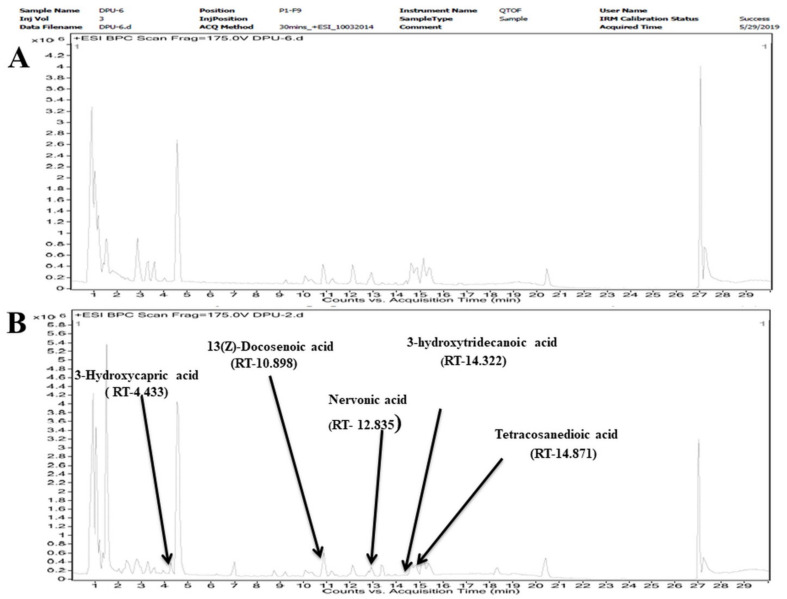
LC-HRMS total ion chromatogram of intracellular metabolites in DMSO- and CUDF-treated MCF-7 cancer cells. (**A**) VTGE-purified elutes of the intracellular metabolite pool in DMSO-treated MCF-7 cells analyzed by LC-HRMS in positive ion mode. (**B**) VTGE-purified elutes of the intracellular metabolite pool of MCF-7 cells treated with CUDF (50 µg/mL) for 72 h analyzed by LC-HRMS in positive ion mode. The detailed procedure is described in the Experimental Procedure section, including the VTGE methodology and the parameters of LC-HRMS.

**Figure 7 biomedicines-11-00889-f007:**
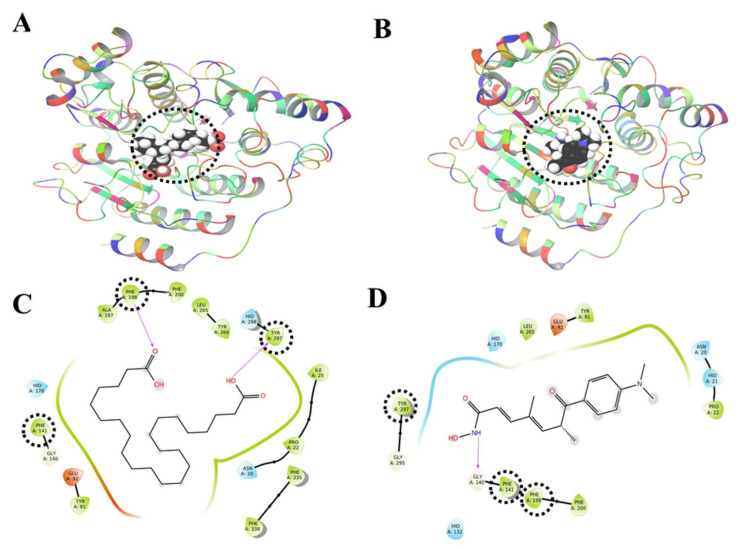
Tetracosanedioic acid, an intracellular long-chain FFA detected in CUDF-treated MCF-7 cells, displays strong molecular interactions at the inhibitory site of HDAC. (**A**) Protein–ligand complex with grey ribbon structure with a full 3D view between HDAC homolog complex with pink color and a biolistic model of tetracosanedioic acid. (**B**) Protein–ligand complex with grey ribbon structure with a full 3D view between HDAC homolog complex with pink color and a biolistic model of trichostatin A. (**C**) Visualizer software-assisted 2D image of the binding mode of tetracosanedioic acid (PubChem CID 2724554) and HDAC homolog complex (1C3R) at the inhibitory site of amino acid residues. (**D**) Visualizer software-assisted 2D image of the binding mode of trichostatin A, a known inhibitor and the HDAC homolog complex (1C3R), at the inhibitory site of amino acid residues.

**Figure 8 biomedicines-11-00889-f008:**
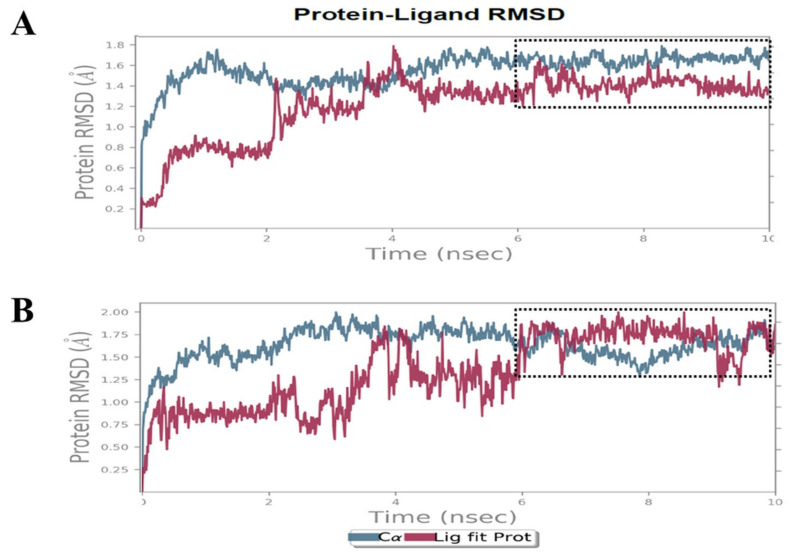
MD simulations predict stable and specific binding to the inhibitory site of the HDAC homolog complex by tetracosanedioic acid and TSA. Root mean square deviation (RMSD) plot of HDAC in complex with tetracosanedioic acid and TSA. The RMSD plot was recorded during 10 ns MD simulations. (**A**) HDAC–tetracosanedioic acid complex; (**B**) HDAC–TSA complex.

**Figure 9 biomedicines-11-00889-f009:**
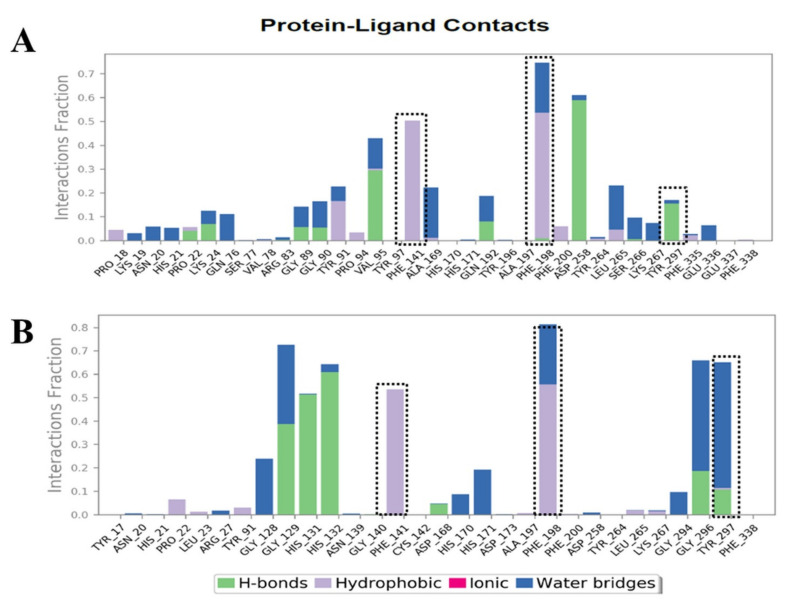
Ligand contact map of tetracosanedioic acid in complex with HDAC projected the inhibitory amino acid residues identical to TSA, a known inhibitor. Protein HDAC–tetracosanedioic acid and TSA complex contact map of interacting amino acid residues generated during 10 ns MD simulations. (**A**) HDAC–tetracosanedioic acid contact map; (**B**) HDAC–TSA, a known inhibitor.

**Figure 10 biomedicines-11-00889-f010:**
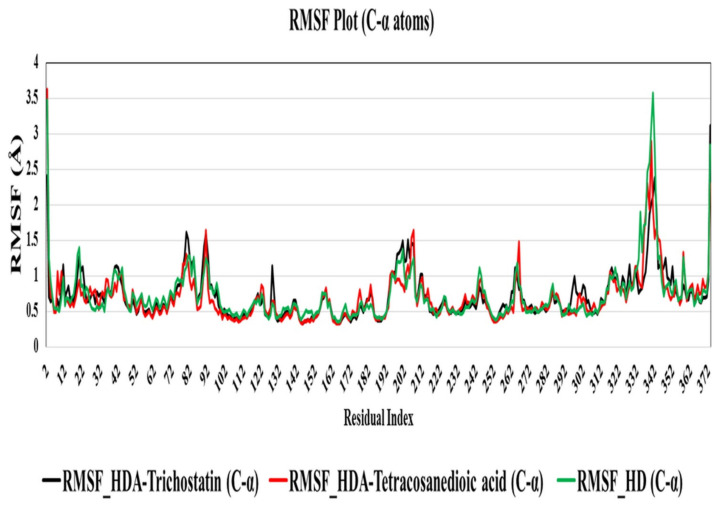
Tetracosanedioic acid and trichostatin A (TSA), a known inhibitor of HDAC, display the least and acceptable fluctuations in complex with HDAC. Root mean square fluctuation (RMSF) plot of HDAC in complex with trichostatin A and tetracosanedioic acid. The RMSF plot was recorded during 10 ns MD simulations. Trichostatin A is shown indicated in black, and tetracosanedioic acid is indicated in red The y-axis represents RMSF values in Angstrom. The RMSF plot of HDAC without any ligand was collected as a control. The x-axis indicates interacting amino acid residues of HDAC with tetracosanedioic acid and TSA.

**Figure 11 biomedicines-11-00889-f011:**
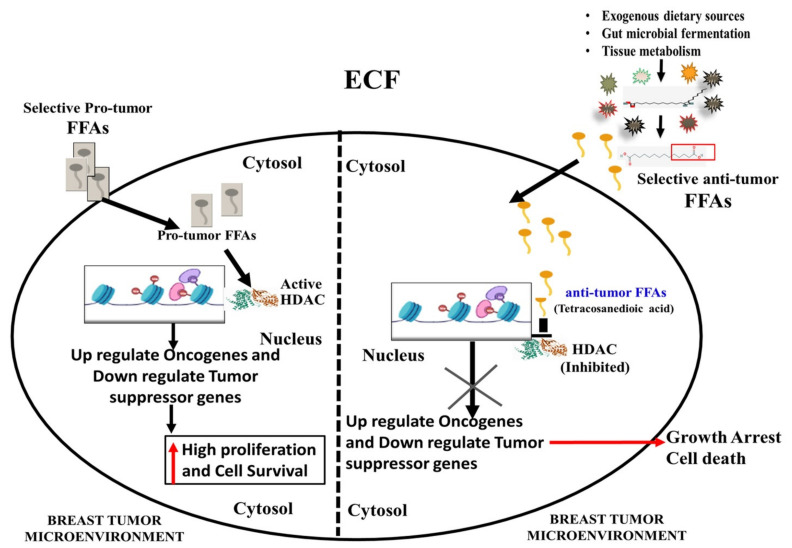
A proposed model of the potential antiproliferative and apoptotic role of FFAs in cancer cells. FFAs, free fatty acids; ECF, extracellular fluid; HDAC, histone deacetylase; FATP, fatty acid translocase protein.

**Table 1 biomedicines-11-00889-t001:** A list of intracellular free fatty acid metabolites in DMSO- and CUDF-treated MCF-7 breast cancer cells. LC-HRMS spectra were obtained in positive ion mode.

Sr. No	Name of Free Fatty Acid	DMSO-Treated MCF-7 Cell Lysate	CUDF-Treated MCF-7 Cell Lysate	RT	*m*/*z*	M.W.
1	13(Z)-Docosenoic acid (monounsaturated fatty acid)	Not detectable	Detectable	10.898	343.2923	338.6
2	Nervonic acid (monounsaturated fatty acids)	Not detectable	45529	12.835	371.3225	317.289
3	Tetracosanedioic acid (Dicarboxylic saturated fatty acid)	Not detectable	Detectable	14.871	399.3537	398.6
4	3-Hydroxytridecanoic acid (Hydroxy saturated fatty acid)	Not detectable	Detectable	14.322	235.1664	230.34
5	8-Hydroxy Caprylic acid (Hydroxy saturated fatty acid)	Not detectable	Detectable	4.433	824.381	160.21

**Table 2 biomedicines-11-00889-t002:** Molecular interactions of intracellular free fatty acids with histone deacetylase (PDB ID 1C3R) show specific binding within the inhibitory sites of enzymes. Binding amino acid residues and the nature of bonds were visualized using DSV-3 on AutoDock Vina docked output complex.

Name of Ligand (PubChem)	Protein PDB ID and Name of the Chain	Binding Affinity (−kcal/mol)	Binding Amino Acid Residues	No. of Bonds	Distance of Bonds
13-Docosenoic acid(monounsaturated fatty acid) (PubChem CID 8216)	1C3R-Histone deacetylase (HDAC)	−7.5	Pro18 His21 Tyr91 Phe141 Ala197 Phe198 Phe200 Leu265	2 H Bonds and 8 Pi Bonds	3.4 3.7 3.2 3.6 4.5 4.7 5.0 6.7
Nervonic acid Monounsaturated (PubChem CID 5281120)	1C3R-Histone deacetylase (HDAC)	−6.5	Tyr91 Tyr196 Phe200 Leu265 Phe335 Phe338	1 H Bond and 5 Pi Bonds	5.0 6.0 4.3 6.5 5.5 5.3
Tetracosanedioic acid (saturated fatty acid) (PubChem CID 119039)	1C3R-Histone deacetylase (HDAC)	−7.0	Tyr297 Leu265 Phe198 Phe141 His132 Pro22	2 H Bonds and 8 Pi Bonds	2.5 2.3 3.7 4.4 3.1
3-Hydroxycapric acid (saturated fatty acid) (PubChem CID 69820)	1C3R-Histone deacetylase (HDAC)	−6.1	Try17 Ala98 Ser103 Ser23 Tyr12 Arg27	5 H Bonds and 8 Pi Bonds	2.1 3.2 4.6 5.2 2.5 4.6
3-Hydroxytridecanoic acid (saturated fatty acid) (PubChem CID 5312749)	1C3R-Histone deacetylase (HDAC)	−6.6	Ile25 Ser29 Arg16 Leu23 Ala98 Ala106	3 H Bonds and 8 Pi Bonds (1 unfavorable)	2.7 3.5 6.4 3.5 4.6 3.8
Trichostatin A(+) (PubChem CID 444732)	1C3R-Histone deacetylase (HDAC)	−9.3	Tyr297 Phe200 Phe198 Gly140 His132 Pro22	2 H Bonds and 8 Pi Bonds	4.0 3.4 3.3 3.2 5.1 5.6

## Data Availability

Not applicable.

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
