# Peer review of "Free Fatty Acids from Cow Urine DMSO Fraction Induce Cell Death in Breast Cancer Cells without Affecting Normal GMSCs"

_biomedicines, 2023, doi:10.3390/biomedicines11030889_

Round 1

Reviewer 1 Report

The manuscript entitled “Free fatty acids from cow urine DMSO fraction induce cell death in breast cancer cells without affecting normal GMSCs” is a good attempt at the subject matter. However, it needs some modifications before it is considered for publication.

1.      The objective of the study is not highlighted in the abstract.

2.      The methodology of section 2.2. (Preparation of CUDF) should be explained in detail. DMSO is a water-miscible solvent. The term number of metabolites and concentration (10 mg/ml) need to be clarified.

3.      Can tetracosanedioic acid be tried as a drug for breast cancer?

4.      What is the complementary fatty acid to tetracosanedioic acid that can be assessed for cancer treatment?

5.      A comparative table of free fatty acids found in COW urine, goat urine, and other ruminants may be provided. It should also be mentioned if these FFA are present in nature, and can be synthesized. or are exclusive to cow/goat etc.

Author Response

Responses to Reviewer 1 Comments:

The manuscript entitled “Free fatty acids from cow urine DMSO fraction induce cell death in breast cancer cells without affecting normal GMSCs” is a good attempt at the subject matter. However, it needs some modifications before it is considered for publication.

Comment 1.      The objective of the study is not highlighted in the abstract.

Response: The authors have included objectives in the abstract.

Comment 2.      The methodology of section 2.2. (Preparation of CUDF) should be explained in detail. DMSO is a water-miscible solvent. The term number of metabolites and concentration (10 mg/ml) need to be clarified.

Response 2: The authors appreciate comments and improved methodology of section 2.2.2. The sentence “The term number of metabolites and concentration (10 mg/ml)” modified to be more appropriate

Comment 3.      Can tetracosanedioic acid be tried as a drug for breast cancer?

Response 3: Then authors propose the potential uses of composition of free fatty acids (COFFAs) including tetracosanedioic acid as a potential drug for breast cancer. In literature, details on tetracosanedioic acid are highly limited, but well correlates with the plant dietary sources, specifically Brassica napus. In fact, Brassica napus is the main dietary source for animals such as cows in the Indian setting.

https://www.sciencedirect.com/science/article/pii/S0022030220306081. https://pubmed.ncbi.nlm.nih.gov/17055542/ https://pubchem.ncbi.nlm.nih.gov/compound/Tetracosanedioic-acid

Comment 4.      What is the complementary fatty acid to tetracosanedioic acid that can be assessed for cancer treatment?

Response 4: The authors have started exploring the potential anticancer effects of composition of free fatty acids (COFFAs) including tetracosanedioic acid, 13Z-docosenoic acid (erucic acid), nervonic acid, 3-hydroxy-tetradecanoic acid, and 3-hydroxycapric acid. Besides anticancer effects, we intend to explore the inhibitory role of these FFAs upon HDAC using in vitro and in silico approaches.

Comment 5.      A comparative table of free fatty acids found in COW urine, goat urine, and other ruminants may be provided. It should also be mentioned if these FFA are present in nature, and can be synthesized. or are exclusive to cow/goat etc.

Response 5: The nature of CUDF derived intracellular FFAs such as tetracosanedioic acid, 13Z-docosenoic acid (erucic acid), nervonic acid, 3-hydroxy-tetradecanoic acid, and 3-hydroxcapric acid are earlier reported in dairy cows fed with diets enriched with forage rape etc.

  • Keim JP, Daza J, Beltrán I, Balocchi OA, Pulido RG, Sepúlveda-Varas P, Pacheco D, Berthiaume R. Milk production responses, rumen fermentation, and blood metabolites of dairy cows fed increasing concentrations of forage rape (Brassica napus ssp. Biennis). J Dairy Sci. 2020 Oct;103(10):9054-9066.
  • Adewuyi AA, Gruys E, van Eerdenburg FJ. Non esterified fatty acids (NEFA) in dairy cattle. A review. Vet Q. 2005 Sep;27(3):117-26.
  • AnitaÅŸ Ö, Göncü S. Relations between feces, urine, milk and blood fatty acid contents in cattle. MOJ Eco Environ Sci. 2018;3(6):356‒362. DOI:10.15406/mojes.2018.03.00113.

Reviewer 2 Report

In the article "Free fatty acids from cow urine DMSO fraction induce cell death in breast cancer cells without affecting normal GMSCs" the authors state that tetracosandioic acid, 13Z-675 docosenoic acid, nervous acid, tetracosandioic acid, 3-hydroxytridecanoic acid and 3-676 hydroxycapric acid are identified at the intracellular level of breast cancer cells, demonstrating arrest of proliferation and induction of apoptotic cell death in MCF-7 and ZR-75-1 breast cancer cells. The reviewer is not entirely clear on the method of extraction of these acids from cultured breast cancer cells. It is not shown whether these cells contained these acids prior to treatment with bovine urine extract. Also, in order to assert that the action of these acids is similar to the effect of trichostatin, it is necessary to try to show the antiproliferative effect of these compounds by means of flow cytometry. What amount of DMSO was contained in the samples of compounds added to the studied breast cancer cells. The effect of these acids on normal cells (e.g., fibroblasts or conditionally normal HEK293. In addition, recently published a series of studies on the antitumor activity of fatty unsaturated acids, acting as effective inhibitors of human topoisomerases [Prog. Lipid Res. 2008, 47, 50–61; Chem. commun. 2013, 49, 8401–8403; Tetrahedron 1997, 53, 16699–16710; Bioorg. Chem. 2020, 104, 104303; Phytochemistry Reviews volume 20, pages 325–342 (2021); Chem Rev 112:3611–3640; Stud Nat Prod Chem 2017, 54:21–86. Doi:10.1016/B978-0-444-63929-5.00002-4; Acc Chem Res 1991, 24:69–75, and other articles from these scientific groups]. I think that the authors of the article also need to keep these studies in mind and discuss them in detail in the article in comparison with their own results.

I believe that after the corrections made and the answers to the questions posed, the article can undoubtedly be accepted for publication, since it is very important and causes a wide multidisciplinary interest.

Author Response

Reviewer Comments 2:

In the article "Free fatty acids from cow urine DMSO fraction induce cell death in breast cancer cells without affecting normal GMSCs" the authors state that tetracosandioic acid, 13Z-675 docosenoic acid, nervous acid, tetracosandioic acid, 3-hydroxytridecanoic acid and 3-676 hydroxycapric acid are identified at the intracellular level of breast cancer cells, demonstrating arrest of proliferation and induction of apoptotic cell death in MCF-7 and ZR-75-1 breast cancer cells.

Comment 1: The reviewer is not entirely clear on the method of extraction of these acids from cultured breast cancer cells.

Response 2: The authors appreciate comment and have improved methodology of section 2.2. (Preparation of CUDF).

Comment 2: It is not shown whether these cells contained these acids prior to treatment with bovine urine extract.

Response 2: The authors appreciate logical comment. The intracellular profiling of MCF-7 cancer cells treated by FFAs enriched CUDF was compared to 10 µl DMSO solvent control. We subtracted the background during profiling and this suggested the FFAs such as tetracosandioic acid, 13Z-675 docosenoic acid, nervous acid, tetracosandioic acid, 3-hydroxytridecanoic acid and 3-676 hydroxycapric acid specific to treated MCF-7 cells.

Comment 3: Also, in order to assert that the action of these acids is similar to the effect of trichostatin, it is necessary to try to show the antiproliferative effect of these compounds by means of flow cytometry.

Response 3: The authors completely agree to the suggestions. In this paper, we have tried to discuss the role of positive control trichostatin A, an inhibitor of HDAC with existing literature. In fact, we have started exploring the anticancer effects of composition of free fatty acids (COFFAs) including tetracosanedioic acid, 13Z-docosenoic acid (erucic acid), nervonic acid, 3-hydroxy-tetradecanoic acid, and 3-hydroxycapric acid. Besides anticancer effects, we intend to explore the inhibitory role of these FFAs upon HDAC using in vitro and in silico approaches. In these experiments, we will consider trichostatin A as a positive control for cell based assays.

Comment 4: What amount of DMSO was contained in the samples of compounds added to the studied breast cancer cells.

Response 4: We have provided details on the preparation of CUDF in DMSO solvent. We have prepared a stock of 10 mg per ml. For 50 µg per ml final concentration of CUDF in treated cancer cells, we added 10 µl of CUDF. Hence, final DMSO concentration in percentage was 0.5% that is within the acceptable range in cell based assays.

Comment 5: The effect of these acids on normal cells (e.g., fibroblasts or conditionally normal HEK293.

Response 5: The authors accept the suggestion. During initial evaluation of CUDF, normal counterpart GMSC cells were available and tested. However, We have started detailed study on anticancer effects of composition of free fatty acids (COFFAs) including tetracosanedioic acid, 13Z-docosenoic acid (erucic acid), nervonic acid, 3-hydroxy-tetradecanoic acid, and 3-hydroxycapric acid and for this, we will include normal cells such as fibroblasts or conditionally normal HEK293.

Comment 6 In addition, recently published a series of studies on the antitumor activity of fatty unsaturated acids, acting as effective inhibitors of human topoisomerases [Prog. Lipid Res. 2008, 47, 50–61; Chem. commun. 2013, 49, 8401–8403; Tetrahedron 1997, 53, 16699–16710; Bioorg. Chem. 2020, 104, 104303; Phytochemistry Reviews volume 20, pages 325–342 (2021); Chem Rev 112:3611–3640; Stud Nat Prod Chem 2017, 54:21–86. Doi:10.1016/B978-0-444-63929-5.00002-4; Acc Chem Res 1991, 24:69–75, and other articles from these scientific groups]. I think that the authors of the article also need to keep these studies in mind and discuss them in detail in the article in comparison with their own results.

Response 6: The authors appreciate suggestions. We have considered these papers and attempted to link with the proposed paper.

I believe that after the corrections made and the answers to the questions posed, the article can undoubtedly be accepted for publication, since it is very important and causes a wide multidisciplinary interest.

Round 2

Reviewer 2 Report

The answers of the authors and the changes made completely satisfied me.